# The Role of Automated Infrared Pupillometry in Traumatic Brain Injury: A Narrative Review

**DOI:** 10.3390/jcm13020614

**Published:** 2024-01-22

**Authors:** Charikleia S. Vrettou, Paraskevi C. Fragkou, Ioannis Mallios, Chrysanthi Barba, Charalambos Giannopoulos, Evdokia Gavrielatou, Ioanna Dimopoulou

**Affiliations:** First Department of Critical Care Medicine & Pulmonary Services, Evangelismos Hospital, Medical School, National and Kapodistrian University of Athens, 10676 Athens, Greeceidimo@otenet.gr (I.D.)

**Keywords:** traumatic brain injury, pupillary light reflex, portable infrared automated pupillometry

## Abstract

Pupillometry, an integral component of neurological examination, serves to evaluate both pupil size and reactivity. The conventional manual assessment exhibits inherent limitations, thereby necessitating the development of portable automated infrared pupillometers (PAIPs). Leveraging infrared technology, these devices provide an objective assessment, proving valuable in the context of brain injury for the detection of neuro-worsening and the facilitation of patient monitoring. In cases of mild brain trauma particularly, traditional methods face constraints. Conversely, in severe brain trauma scenarios, PAIPs contribute to neuro-prognostication and non-invasive neuromonitoring. Parameters derived from PAIPs exhibit correlations with changes in intracranial pressure. It is important to acknowledge, however, that PAIPs cannot replace invasive intracranial pressure monitoring while their widespread adoption awaits robust support from clinical studies. Ongoing research endeavors delve into the role of PAIPs in managing critical neuro-worsening in brain trauma patients, underscoring the non-invasive monitoring advantages while emphasizing the imperative for further clinical validation. Future advancements in this domain encompass sophisticated pupillary assessment tools and the integration of smartphone applications, emblematic of a continually evolving landscape.

## 1. Introduction

Pupillometry refers to the examination of the pupil of the eye, by measuring the pupil size and its reactivity, and is an essential component of the neurological examination. The pupillary reaction is the output of several neural reflexes, namely, the pupillary light reflex (PLR), the pupillary dark reflex [1,2], the ciliospinal reflex [3,4], and the near accommodation reflex [1,2,4] (see also Table 1). The PLR involves the narrowing of the pupil when exposed to light, thus regulating the quantity of light entering the retina [2], and is of particular importance because of the ease of its assessment, even in patients who are in comatose states [5]. The afferent fibers of the reflex originate from the retinal ganglion cell layer in the eye and proceed along the optic nerve, optic chiasm, and optic tract to connect with the brachium of the superior colliculus in the midbrain. The tract continues to the pretectal area of the midbrain, sending fibers bilaterally to the efferent Edinger–Westphal nuclei. From the Edinger–Westphal nucleus, the preganglionic parasympathetic fibers responsible for pupillary constriction travel through the oculomotor nerve to reach the ciliary ganglion, where they synapse. Postganglionic parasympathetic axons are conveyed through the short ciliary nerve to stimulate the iris sphincter. The innervation of both Edinger–Westphal nuclei results in the generation of a direct and consensual pupillary response (Figure 1). The structural integrity of the afferent and efferent routes that are located in the brain and the brainstem is theoretically necessary for a normal PLR. The pupil size, dilation, and constriction velocities are also influenced by the autonomic nervous system, with the sympathetic nervous system tending to dilate the pupil (mydriasis) and the parasympathetic nervous system tending to constrict the pupil (myosis) [6] (Figure 1).

Traditionally, assessing pupils involved a pupil gauge for estimating size and a penlight or flashlight for manual evaluation of reactivity, categorized as “normal”, “brisk”, “slow”, or “sluggish”. In a recent study by Couret and colleagues, they found limited consistency between measurements of the pupil size obtained through an automated device compared to traditional clinical evaluation, revealing an overall disagreement rate of 18%. When the pupil size was less than 2 mm, the rate of disagreement escalated to 39%, demonstrating that conventional methods of assessing the PLR may provide inaccurate information [7].

With recent technological advancements, portable automated infrared pupillometers (PAIPs) have emerged, offering a user-friendly solution without the need for specialized training. These devices facilitate an objective assessment of the PLR by measuring pupil size, constriction, and dilation velocities [8]. These tools extend beyond ophthalmology cabinets, finding applications in clinical and research settings. Key areas include emergency departments for brain injury assessment [9,10,11,12], sports settings [13,14,15], for early detection of neuro-worsening [16,17,18], non-invasive monitoring of intracranial hypertension in critically ill brain trauma patients [19,20,21,22], post-cardiac-arrest evaluations of brainstem function [5,23,24,25], and even in cases involving extracorporeal cardiopulmonary resuscitation [26]. Notably, during cardiopulmonary resuscitation (CPR), PLR serves as a surrogate method for monitoring CPR effectiveness and may predict the likelihood of returning to spontaneous circulation as evidenced in a cardiac arrest swine model [27].

Applications of PAIPs extend to assessing clinical conditions impacting the autonomic nervous system [28,29,30,31], neurodegenerative disorders [32,33,34], and psychiatric conditions, such as autism, recreational drug use, and alcohol consumption [35,36]. In this narrative review, we focus mainly on the use of PAIPs in patients with traumatic brain injury (TBI). We present the current evidence on the use of these devices for the diagnosis, monitoring, and prognostication of mild, moderate, and severe TBI and attempt to explore their strengths and limitations when applied in modern intensive care units (ICUs) and emergency departments (EDs).

## 2. Methods

To compose this article, we drew upon the expanding literature concerning the quantitative assessment of the PLR and our professional experience derived from applying PAIPs in our department. We aimed to comply with the Scale for the Assessment of Narrative Review Articles (SANRA) [37]. The primary objective of our work is to provide clinicians with insights into the utility of PAIPs in the care of TBI patients and to explore the advantages and limitations of these tools. Given our role as ICU clinicians, our patient demographic predominantly comprises individuals with severe TBI (sTBI). Acknowledging the distinct characteristics and needs of this population compared to those with mild TBI (mTBI), who are infrequently attended to by intensivists but often by other practitioners and neurosurgeons, we opted to delineate mTBI and sTBI in two separate sections in our manuscript. We conducted a literature review, focusing primarily on the last decade, utilizing databases, such as PubMed and Google Scholar. Employing keywords such as “pupillometry” and “traumatic brain injury”, we aimed to identify the pertinent articles. Our prioritization criteria encompassed prospective multicenter studies, systematic reviews, and general multicenter studies. Additionally, we considered recent descriptive reviews that presented technical or physiological concepts exceptionally. Case reports were incorporated where they provided insights into aspects of the PLR examination not covered in the already selected publications. In the process of article prioritization, we favored prospective studies over retrospective ones, particularly if the latter were conducted at a single center. Each author proposed articles for inclusion, and in instances of disagreement, consensus was achieved through discussion.

## 3. Technical Characteristics of PAIPs

PAIP devices are portable, handheld instruments that operate independently. Typically, they come equipped with a color Liquid Crystal Display (LCD), an integrated digital camera, and a built-in light source. Infrared light with a wavelength of 850 nm, outside the typical range of human eye sensitivity, is used to illuminate the patient’s eye. Images are captured using a digital camera equipped with an infrared-sensitive sensor array. Each measurement sequence involves analyzing over 100 images, providing detailed information on various pupillary functions within a rapid 3 to 4 s timeframe. PAIPs are specifically designed to minimize risks to both users and patients. The sole point of mechanical contact with the patient is usually the headrest of the device. Radiation levels consistently remain below the threshold values recommended by the International Commission on Non-Ionizing Radiation Protection. In a worst-case scenario, infrared exposure is restricted to 30 times below the maximum permissible exposure, and visible radiation stays well within the safety limits. PAIPs demonstrate detection thresholds for changes in pupil size of less than 0.05 mm [8,38].

While diverse PAIP devices yield comparable values for pupil size and constriction to light stimulation, there is evidence supporting variability, particularly in pupillary latency values among different devices. Importantly, absolute values of measured variables are not interchangeable, and this is likely attributable to distinct technical characteristics inherent to each device. Discrepancies may also arise due to variations in eye coverage during procedures, with some devices offering coverage during measurements, while others do not. An optimal PLR assessment may, therefore, involve conducting measurements in a darker environment. Furthermore, certain devices provide additional calculated parameters, such as the Neurological Pupil Index (NPi) (NeurOptics, Laguna Hills, CA, USA). Calculated by a proprietary algorithm using multiple variables (latency, constriction velocity, pupil size at baseline, % change, and dilation velocity), an NPi value below 3.0 is generally considered abnormal based on normative data [39].

## 4. The Normal PLR Response

Distinct from the conventional approach, PAIPs deliver a quantitative assessment alongside a PLR curve, as illustrated in Figure 2, thereby enabling an objective and consistent assessment of PLR parameters. Commonly measured parameters include pupil size before light stimulus, latency (the time from light stimulus to the first constriction), constriction velocity (the speed of pupil constriction), and dilation velocity (the speed of pupil recovery to pre-stimulus size) [39]. Despite the availability of normative data, it is noteworthy that measured values may exhibit variations when different PAIP devices are employed. Table 2 presents some indicative pupillometry values obtained with a PAIP from healthy volunteers [38] and from a multicenter registry including neurocritical care patients [39]. It has been described that in healthy volunteers, the initial pupil diameter is directly related to the constriction velocity, and even though the initial pupil diameter varies due to fatigue and the ambient light level, a constant positive correlation has also been observed between the initial and post-stimulus diameter [38]. Pupillary asymmetry measuring < 0.5 mm (which would be difficult to detect with traditional PLR assessment) has been observed in normal subjects, indicating that this asymmetry is within the normal range [38]. Recently, the evaluation of pupil size and reactivity has been embedded into the Glasgow Coma Scale (GCS) assessment through the creation of the GCS-Pupil score (GCS-P), in which the number of non-reacting pupils is subtracted from the total GCS score [40].

## 5. Evidence for PAIP Application in mTBI

MTBI is characterized by a GCS score ranging from 13 to 15, recorded 30 min or later post-injury, [41], accompanied by a history of brief or no loss of consciousness and a short period of post-traumatic amnesia. Traditionally, minimal clinical intervention has been deemed sufficient for mTBI, primarily involving watchful waiting [42]. While computed tomography (CT) may reveal macrostructural intracranial injuries in some cases [43], advanced imaging techniques such as magnetic resonance diffusion tensor imaging have illuminated potential microstructural pathology [44,45]. The clinical significance of these changes, however, remains elusive. Routine acute neuroimaging is not universally recommended in mTBI. Individuals aged 65 and above and patients on anticoagulation therapy face an increased risk of intracranial bleeding, and for these reasons, it is advisable to regularly perform CT scans and/or consider hospital admission for observation in these populations. It has already been acknowledged, however, that the standardized evaluation of both the level of consciousness and of pupillary reactivity are basic components of the clinical evaluation of all patients with brain trauma, and therefore, PAIP use could in theory align well with the proposed standards for examination and treatments for the hospitals that offer a neurotrauma service [42,46].

Diagnosing mTBI necessitates, as a crucial step, establishing a plausible injury mechanism and, subsequently, assessing the patient for signs and symptoms of altered mental status. Confounding factors, such as substance intoxication, may, however, cause difficulties in the clinical assessment [42,47,48]. Establishing appropriate biomarkers is therefore a research priority in mTBI. PAIP devices could be used in the triage of such patients in theory, but clinical studies supporting this use of PAIPs are lacking, and most research publications investigating PLR changes in mTBI include patients in the subacute and chronic phases.

For example, Master et al. [14] performed an observational study including healthy control individuals and athletes with a diagnosis of sport-related concussion within 28 days post-injury. They measured the PLR parameters under moderate photopic conditions (approximately 350 lux), and they found the pupil sizes and PLR velocities to be significantly greater among the athletes with concussions compared to the healthy controls. In exploratory analyses, females with concussion also exhibited longer T75. Other researchers, however, report different results. In a study involving individuals 15–45 days after injury, where the PLR was assessed under mesopic conditions (3 lux), the mTBI group had increased constriction latency, suggesting a slower reaction, compared to the control group. Similarly, the mTBI group exhibited a reduced average constriction velocity and a slower dilation velocity and time needed for the pupil to reach 75% of its original size [49]. Accordingly, Thiagarajan and colleagues, who examined the PLR under photopic conditions (350 lux) in individuals with non-blast-induced, chronic mTBI, described significant reductions in the constriction and dilation velocities and pupil diameters in mTBI cases compared to the controls. These findings led the authors to the conclusion that the slowed dynamics and decreased pupillary diameters in mTBI indicate a deficiency primarily in the sympathetic system and more subtle parasympathetic involvement [50]. Finally, Truong et al. also reported smaller pupil diameters and symmetrical, slower pupillary dynamics [51] in mTBI patients [52]. Among mTBI patients, subjects with photosensitivity were found to exhibit larger pupil diameters and faster PLR dynamics [53].

## 6. Evidence for PAIP Application in Moderate and sTBI

The existing management algorithms for moderate (GCS 9–12) and sTBI, i.e., TBI with a GCS < 9, include the PLR assessment both as an essential component in the neurological examination and as a prognostic indicator [54,55,56]. The loss of the PLR has repeatedly been identified as a marker of poor prognosis in patients with sTBI, including those receiving emergency neurosurgery [57]. Even in the setting of sTBI management without invasive intracranial pressure (ICP) monitoring, a PLR assessment is included as a criterion to initiate and escalate or de-escalate treatment [55]. Currently, and particularly in low-income settings, pupillary reactivity is examined by clinicians using manual flash penlights. However, clinicians show a preference for using PAIPs rather than the conventional flashlight method for PLR examination [58], which is non-quantitative and inter-operator-dependent [59]. Using PAIPs in low-income settings can facilitate the management of sTBI patients by reducing errors in PLR assessment [59,60].

An important concept related to the management of TBI in both the ED and ICU settings is that of critical neuro-worsening. Critical neuro-worsening refers to a serious deterioration in clinical neurologic status that warrants immediate action to reverse the threat to the brain. In the context of TBI, critical neuro-worsening includes a spontaneous decrease in the motor GCS by ≥1 point compared to the previous examination, evolution of a new focal motor deficit, herniation syndrome or Cushing’s triad, and a new decrease in pupillary reactivity or new pupillary asymmetry or bilateral mydriasis (see also Figure 3). The immediate response to critical neuro-worsening requires the emergent evaluation to identify possible causes, the consideration of emergent imaging, and the rapid escalation of treatment, including hyperosmolar therapy and acute hyperventilation, if herniation is suspected [54,55,56]. It follows that accurate and repeated evaluation and documentation of pupillary reactivity is important to identify critical neuro-worsening in a timely and accurate manner and avoid both under- and over-treatment, which both carry significant risks for patients with TBI [56]. In this direction, there is accumulating evidence for a relationship between the NPi and ICP, with values <3 being indicative of increased ICP [21].

The quantitative PLR assessment particularly within the initial three days of hospitalization for individuals with brain trauma has emerged as a predictor of clinical outcomes when integrated with demographic and imaging data [61]. Consistent evidence points to persistently low NPi values (<3) being associated with intricate ICP dynamics and overall poorer outcomes [62]. Significantly, the response of PLR measurements to ICP-lowering treatments has shown a promising connection with improved outcomes [38,62,63,64,65].

Singer et al. describe significant differences in PLR measurements between patients with mild and those with sTBI. However, these observed values did not exhibit a direct correlation with ICP, a phenomenon potentially attributed to the inclusion of patients with mTBI in the study [66]. In instances of focal pathology, a negative correlation between ICP and the NPi has been discerned [62]. Moreover, patients with ICP exceeding 35 mmHg showed slower constriction velocities and smaller pupillary diameters, as detailed in the research by Chen et al. [67].

Delving into the potential application of the NPi as a non-invasive tool for assessing ICP, Pansell et al. report an Area Under the Curve (AUC) of 0.72, with an optimal cutoff value above 3.9 for effectively ruling out intracranial hypertension [68]. A notable observation from a distinct study unveils that patients with pathological PLR showcase initial abnormal measurements approximately 16 h before the peak of the observed ICP [67].

The ORANGE study represents the most extensive prospective observational cohort investigation devised to evaluate the prognostic significance of multiple NPi measurements over time, adjusting for various established baseline predictors. Within this study, consistently abnormal NPi values, encompassing the most extreme values reaching 0, within the first week after acute brain injury, were indicative of unfavorable outcomes. Moreover, two consecutive NPi measurements equal to 0 or a deterioration of the NPi to a value of 0 were linked to an elevated mortality risk. In contrast, the risk of mortality did not increase when an NPi value of 0 recovered to a higher value. Notably, NPi values falling between 3 and 4 demonstrated a significantly higher association with mortality risk compared to NPi values exceeding 4 [69]. These findings propose that therapeutic interventions may potentially reverse brainstem injuries, leading to subsequent PLR and clinical improvement.

Although there is growing evidence supporting the use of a PAIP for neuro-prognostication and clinical management recommendations, invasive ICP monitoring cannot be replaced by PAIP-derived measurements. In a prospective study conducted by Robba et al., including adult ICU patients with sTBI in whom invasive ICP monitoring had been initiated, estimates for the optic nerve sheath diameter, pulsatility index, non-invasive ICP using transcranial Doppler, and NPi were simultaneously collected. The authors found that the most accurate non-invasive method for estimating the existence of ICH was to combine the measurement of the optic nerve sheath diameter with the non-invasive ICP obtained by transcranial Doppler [21].

## 7. Limitations of Pupillometry in TBI Assessment

The monitoring of TBI patients with PAIPs can be highly beneficial in clinical practice, as an alternative to invasive methods, thus reducing the associated risks. Nevertheless, it comes with significant limitations [70]. Pupillary changes can stem from various reasons unrelated to neurological conditions, such as medications, emotions, or systemic illnesses, making it less specific for TBI. Pupillometry does not offer direct insight into the underlying cause of PLR derangements. Additional assessments, testing, and clinical judgment are essential to determine the root cause of observed changes. Pupillometry captures data at a specific point, potentially missing dynamic trends in neurological status over time unless consistently repeated. The accuracy of PAIP measurements can be influenced by the operator’s skill and experience, with inexperienced operators potentially introducing errors into the measurements. Accurate measurements ideally require controlled lighting conditions, as changes in ambient light levels can impact pupil size and reactivity, potentially prompting misleading results [71,72]. In cases of mTBI or instances with subtle neurological changes, pupillary responses may not display significant alterations, limiting its diagnostic value. Finally, pupillary responses can vary widely among individuals, underscoring the importance of establishing a baseline for each patient.

Certain medications, particularly those impacting the autonomic nervous system, can influence pupillary size and reactivity [73]. This consideration is vital when interpreting pupillometry results. Several drugs commonly utilized in ICUs have been examined to understand their impact on the PLR. For instance, opioids are known to induce miosis, but they do not seem to significantly affect other PLR parameters [74,75]. Benzodiazepines have no notable effect on the PLR, while propofol does induce a significant reduction in both pupil diameter and constriction velocities. Although the impact of barbiturates on the PLR has been less extensively studied, it is known that during burst suppression, pupillary constriction velocities can be reduced to <0.6 mm/s and constriction to <10% of the baseline size. Moreover, the ciliospinal reflex, which involves pupil dilation in response to a painful stimulus in the head or neck, may be present or exaggerated during a barbiturate coma [34,70,71,72]. Catecholamines may diminish the amplitude of the PLR [76], while paralytic agents do not have a significant effect [76].

Commercially available PAIPs lack the capability for bilateral observation, rendering the assessment of relative afferent defects impossible. Furthermore, the use of currently available commercial PAIPs with fixed light intensity and duration may affect the sensitivity of the examination. Finally, agitated or confused patients as well as patients with scleral or periorbital edema, intraocular lens replacement, or prior ocular surgical procedures can be difficult to evaluate [77,78]. Ultrasound pupillometry can be applied in some cases [79] of facial or orbital trauma, but there are currently no automated portable devices offering standardized ultrasound measurements (Figure 4).

## 8. Discussion

Regarding the application of PAIPs in mTBI, there are different metrics reported in different studies. These can be attributed to the use of different PAIP devices but also to the different study populations, including patients at various time points following the mTBI. Interestingly, there is a paucity of data on the use of PAIPs in mTBI patients in the ED. There are currently no defined algorithms to guide emergency clinicians on how they could incorporate PAIP use in the treatment algorithms, even though this could be a promising course. In the meantime, PAIPs can support the accurate and prompt evaluation of neuro-worsening in patients with more severe clinical presentations. Practitioners should be aware that around 20% of individuals exhibit a minor discrepancy in pupil size, a condition referred to as physiological anisocoria [80]. When using PAIP devices, clinicians should evaluate the measured parameters in both eyes, as well as the produced PLR curves (see also Figure 2). A proposed methodology for a structured approach in the ICU setting is the following: First, assess the presence of anisocoria, keeping in mind that the presence of new or >1 mm anisocoria is more likely to be pathological. Next, examine whether there is a pupillary reaction to light stimulus. Some reduction in pupil size is expected with illumination, and the pupillometry curve should not be flat; the NPi (if available) should be > 0. Subsequently, determine if the PLR parameters are within the normal range. For this purpose, the normative data from Table 2 can be used, and, according to the ORANGE study, an NPi value ≥ 4 is associated with a better prognosis. Finally, assess whether the PLR is symmetrical by observing if the pupillometry curves are similar and if the NPi (if available) is equal in both eyes. A difference in the NPi > 0.7 between eyes is currently considered abnormal. Midbrain dysfunction usually produces a characteristic pattern in automated pupillometry measurements (Figure 3). Physiological phenomena that can interfere with PLR measurements include hippus or pupillary unrest [81,82] (see also Figure 5) as well as other neurological [83] or non-structural causes of anisocoria [73].

The ORANGE study marked a significant advancement in utilizing and interpreting NPi values for patients with sTBI. While it was previously established that the absence of PLR, corresponding to an NPi = 0, indicated a poor prognosis, the “normal” NPi limit was conventionally set at 3 based on normative data. However, the ORANGE study, incorporating sTBI patients with repeated measurements, revealed that NPi values between 3 and 4 were also associated with an elevated risk of mortality. In this context, the assessment of the NPi can serve as a tool to identify high-risk patients who might benefit from intensified monitoring and early interventions to prevent secondary neurological complications, allowing for more aggressive therapeutic approaches before irreversible brain damage ensues. Perhaps the most pivotal takeaway from the ORANGE study is the emphasis on evaluating repeated measurements rather than relying solely on isolated readings from modern PAIPs. Clinicians should recognize that a single abnormal PLR measurement should prompt a thorough re-examination of the patient to better understand their condition and make appropriate therapeutic plans.

Despite the gathering evidence supporting the use of PAIPs in sTBI, many studies focus on the NPi and do not provide data on other PLR parameters. This is an important caveat, because not all PAIPs have the possibility to calculate the NPi, which is a NeurOptics patent. Moreover, there is little information on the symmetry of PLR measurements in most of the described studies, which is a limitation, given the importance of anisocoria in traditional PLR assessment. While PAIPs may be more effective than a clinical pupillary assessment in detecting brain herniation or identifying neuro-worsening, further research is needed to establish optimal cutoffs and thresholds for different devices. Physicians should be mindful of device differences in evaluating pupil size, constriction, and latency, as these parameters are not interchangeable.

## 9. Conclusions

A PLR assessment with PAIP devices provides a quantitative analysis of the pupillary parameters and signals a shift from traditional methods of estimating pupillary reactivity. In mTBI, PAIPs offer the potential for triage and monitoring, but clinical studies supporting their use are lacking, while diverse technical characteristics of different PAIP devices influence the repeatability and comparability of measurements between studies.

In sTBI, PAIPs align with existing management protocols and may serve as non-invasive neuro-monitoring tools. The NPi emerges as a potential prognostic biomarker, correlating with elevated intracranial pressure and predicting outcomes. Despite their utility, PAIPs do not replace invasive monitoring and should be regarded as complementary tools. The limitations of PAIP use include non-specific changes influenced by medications and the necessity for controlled conditions during measurements; consequently, their application in TBI necessitates further validation through robust clinical studies. As technology evolves, PAIPs may become integral in the comprehensive evaluation and management of brain trauma.

## 10. Future Directions

Advancements in future hardware technology and associated computer algorithms are poised to yield newer, more cost-effective, and heightened sensitivity instrumentation for pupillary assessment. For instance, upcoming portable pupillometers are anticipated to integrate additional stimuli, such as red light and step profiles, enhancing and broadening their diagnostic capabilities [84]. The development of a simple, affordable, handheld pupillometer capable of accurate binocular recording, monocular/binocular light stimulation, and automated analysis and display is also likely. This development is expected as the medical and related professional communities, including optometry and neurology, increasingly recognize its value [84]. A promising and cost-effective alternative to PAIPs is the emergence of smartphone-based software applications. Preliminary studies indicate comparable results between these smartphone applications and traditional infrared pupillometry [85,86].

Another field that could benefit from the application of PAIPs could be the intra- and perioperative management of neurological surgery with awake patients. Pain can affect PAIP measurements, and monitoring of the PLR has already been applied for the assessment of sedation depth and adequacy of analgesia in the perioperative setting [87]. However, the method could have additional benefits for neurological cases. In a proof-of-concept study, Isnardon and colleagues studied patients with unilateral sciatic nerve block, who were also administered intravenous opioids for perioperative analgesia, and found that the pupillary dilation response to a noxious stimulus was blunted when the blocked side was tested [88]. The use of PAIPs during awake craniotomy could, in theory, provide information not only on analgo-sedation adequacy but also on the functional integrity of the PLR pathway; however, there are not any commercially available PAIPs designed for this purpose at the moment.

Finally, PAIPs have recently been proposed as a surrogate tool for detecting consciousness in patients with acute disorders of both traumatic and non-traumatic etiology [89]. This innovative application of PAIPs could significantly impact the prognostication of sTBI patients, particularly those who emerge from a coma, and support the design of novel treatments.

## Figures and Tables

**Figure 1 jcm-13-00614-f001:**
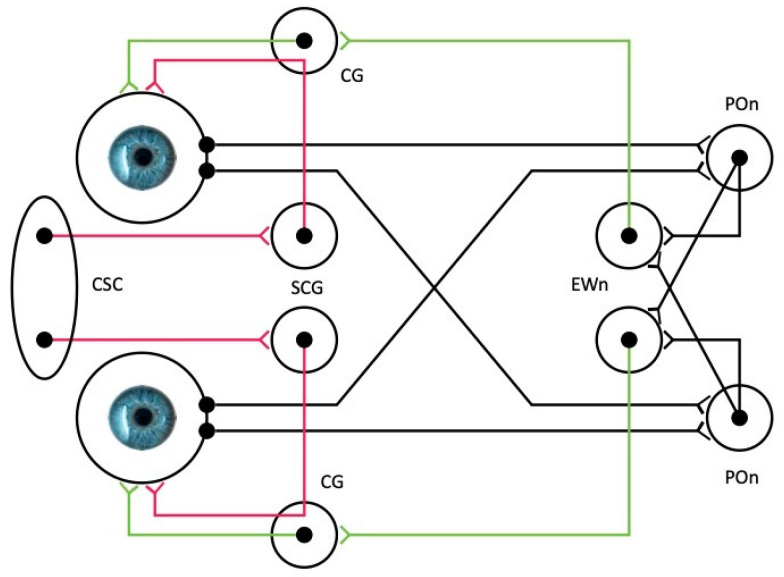
Simplified schematic illustration of the autonomic innervation of the iris. The afferent pathway is shown in black, the parasympathetic pathway in green, and the sympathetic in red. Sympathetic postganglionic neurons may also bypass the ciliary ganglion (pathways not shown). The parasympathetic pathway is shared by both the pupillary light reflex (sphincter pupillae) and the near accommodation reflex (ciliary muscle). EWn, Edinger–Westphal nucleus; POn, pretectal olivary nucleus; CG, ciliary ganglion; SCG, superior cervical ganglion; and CSC, ciliospinal center (C8-T1 segmental level of the spinal cord).

**Figure 2 jcm-13-00614-f002:**
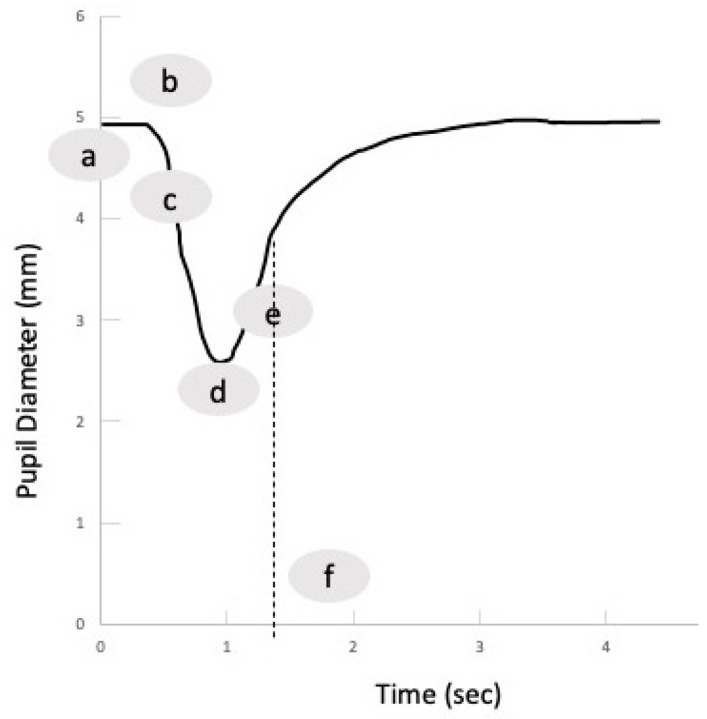
The normal reaction of the pupil to a light stimulus: a, maximum (baseline) pupil size; b, latency period; c, constriction phase; d, minimum size; e, dilation phase; and f, T75, i.e., time in which the pupil recovers 75% of its baseline size (dashed line).

**Figure 3 jcm-13-00614-f003:**
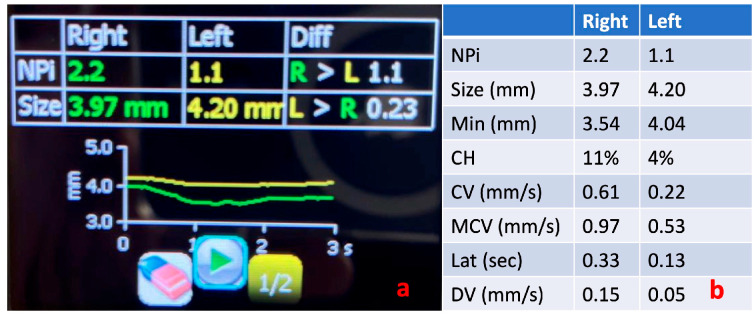
Pupillometry evaluation of a patient with brainstem dysfunction and acute deterioration of the level of consciousness. The green line shows the direct pupillary response of the right pupil to a flashlight stimulus and the yellow line shows the response of the left pupil. Notice the flattening of the left eye pupillometry curve compared to the right (**a**) and the difference in the NPi value between eyes (**b**). NPi, Neurological Pupil Index; Min, minimum pupil size; CH, % constriction; CV, constriction velocity (mean); MCV, maximum constriction velocity; Lat, latency; and DV, dilation velocity.

**Figure 4 jcm-13-00614-f004:**
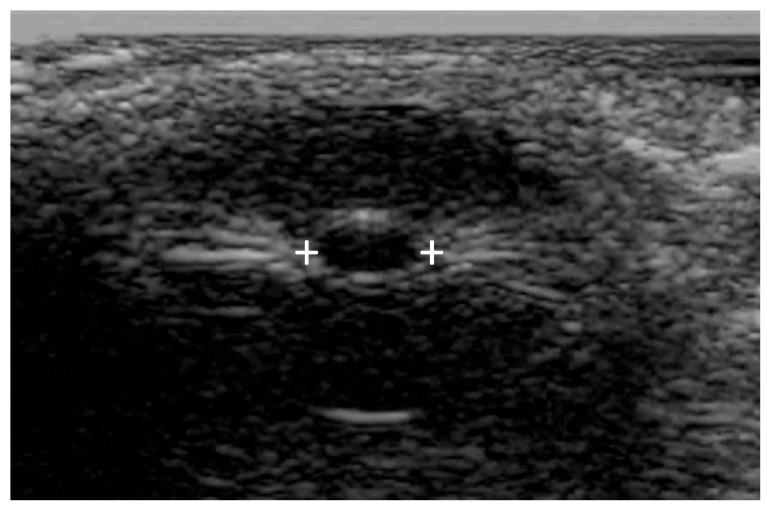
Bedside or portable ultrasound can be used to assess the pupillary reactivity when one or both eyes cannot be opened. Pupillary size is indicated by the plus signs.

**Figure 5 jcm-13-00614-f005:**
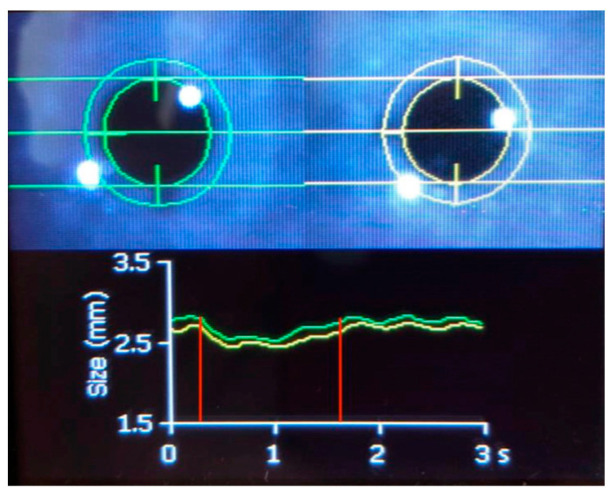
Automated pupillometry in a 30-year-old woman who was intubated and mechanically ventilated with coma and normal brain imaging. The green line shows the direct pupillary response of the right pupil to a flashlight stimulus and the yellow line shows the response of the left pupil. The pupillary constriction to light (shown in the area between the red vertical lines) is preserved, and an oscillation of approximately 1.5 Hz is superimposed. This phenomenon is hippus [81]. Hippus can be normal or pathological. Hippus comes from the Greek “hippos”, meaning horse, and refers to rhythmic, dilating, and contracting pupillary movements [82]. Hippus has been related to epilepsy; however, in this case, concomitant electroencephalography did not detect epileptic activity.

**Table 1 jcm-13-00614-t001:** Key functions of pupillary reflexes.

Pupillary Reflex	Function
Pupillary light reflex	Pupillary constriction to light
Pupillary dark reflex	Pupillary dilation in darkness
Ciliospinal reflex	Pupillary dilation in response to noxious stimuli to the face, neck, and upper trunk.
Near accommodation reflex	Pupil and lens accommodation and convergence of the eyes for near vision

**Table 2 jcm-13-00614-t002:** Reported values for commonly measured pupillometry parameters [38,39].

Pupillometry Parameter *	Normal Volunteers	Neurocritical Care Patients
Maximum pupil size (mm)	4.1 ± 0.34	3.5 ± 1.2
Minimum pupil size (mm)	2.7 ± 0.21	-
Mean reduction in size %	34	-
Mean constriction velocity (mm/sec)	1.48 ± 0.33	1.6 ± 0.9
Mean latency duration (sec)	0.24 ± 0.4	0.3 ± 0.1

* Mean ± standard deviation unless indicated otherwise.

## Data Availability

Not applicable.

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
