# Peer review of "The Role of Automated Infrared Pupillometry in Traumatic Brain Injury: A Narrative Review"

_jcm, 2024, doi:10.3390/jcm13020614_

Round 1

Reviewer 1 Report

Comments and Suggestions for Authors

Interesting work and I congratulate the authors for describing and reviewing articles about the technique that presents great perspectives for assistance in the non-invasive neuromonitoring of neurocritical patients. They carried out an extensive review on the topic with important points covered. I have some questions and suggestions for the authors. What are the implications and advantages that the use of PAIPs could have during the perioperative period of neurological surgery with awake patients? The authors could address this issue in one paragraph.

Reviewer 2 Report

Comments and Suggestions for Authors

I want to congratulate the authors on an excellent narrative review covering the anatomical and physiological basis of portable automated infrared pupillometry (PAIP) in traumatic brain injury (TBI). The article stands out for the clarity of the background provided and the focus on mTBI vs moderate and severe TBI. I enjoyed reading it and would recommend the following corrections to the  authors to further improve the quality of their work: 

- Introduction: Perhaps a table summarising the key principles of the pupillary light reflex,  pupillary dark reflex, the ciliospinal reflex, and the near accommodation reflex would enrich this article and make it even more educational for our readership.

- Methods: I would suggest creating a Methodology section after Introduction and list the process for identifications of the themes covered in this review and how the articles discussed were selected (you could mention which author/s were involved in triaging the articles and how disagreements were addressed). I would recommend making use and referencing the SANRA guidelines for reporting narrative reviews (Baethge C, Goldbeck-Wood S, Mertens S. SANRA-a scale for the quality assessment of narrative review articles. Res Integr Peer Rev. 2019 Mar 26;4:5. doi: 10.1186/s41073-019-0064-8.).

- The normal PLR response: at this end of this paragraph I would mention that the evaluation of pupils size and reactivity has been recently embedded into the GCS assessment through the creation of the GCS-P this will introduce the following sections on mild and moderate/severe TBI (Brennan PM, Murray GD, Teasdale GM. Simplifying the use of prognostic information in traumatic brain injury. Part 1: The GCS-Pupils score: an extended index of clinical severity. J Neurosurg. 2018;128(6):1612-1620. doi: 10.3171/2017.12.JNS172780.).

- Evidence for PAIP in mild TBI: I would stress the importance of standardising clinical practice through PAIP, this is an aspect underscored by Dasic et al. in their scoping review on neurotrauma service in major trauma centres and I would suggest to highlight how PAIP align well with the matrix proposed by those authors regarding aspects of prognostication, best practices, monitoring and follow up of mild TBI (Dasic D, Morgan L, Panezai A, Syrmos N, Ligarotti GKI, Zaed I, et al. A scoping review on the challenges, improvement programs, and relevant output metrics for neurotrauma services in major trauma centers. Surg Neurol Int. 2022;13:171. doi: 10.25259/SNI_203_2022.)

- Evidence of PAIP in moderate and severe TBI: The use of PAIP and the ability to spot early changes in pupils size and reactivity is know to anticipate the neurological worsening in patients with severe TBI and this aspect should be highlighted even more in this paragraph (Clark D, Joannides A, Adeleye AO, Bajamal AH, Bashford T, Biluts H, et al. Casemix, management, and mortality of patients rreseceiving emergency neurosurgery for traumatic brain injury in the Global Neurotrauma Outcomes Study: a prospective observational cohort study. Lancet Neurol. 2022;21(5):438-449. doi: 10.1016/S1474-4422(22)00037-0.)

- Minor comments: Line 290, please rephrase as "potentially prompting misleading results"; Figures 3 and 5 please ensure to increse their quality at least to 600dpi.

Once again, many thanks for the opportunity to read your manuscript, I hope my suggestions will assist you in the revision process. I look forward to receive your revised file.  

Round 2

Reviewer 2 Report

Comments and Suggestions for Authors

Authors have revised appropriately